# Evolution of Recrystallized Grain and Texture of Cold-Drawn Pure Mg Wire and Their Effect on Mechanical Properties

**DOI:** 10.3390/ma13020427

**Published:** 2020-01-16

**Authors:** Jiao Meng, Liuxia Sun, Yue Zhang, Feng Xue, Chenglin Chu, Jing Bai

**Affiliations:** 1School of Materials Science and Engineering, Southeast University, Nanjing 211189, China; joejoe1st@yeah.net (J.M.); zy_seu@foxmail.com (Y.Z.); xuefeng@seu.edu.cn (F.X.); clchu@seu.edu.cn (C.C.); 2Jiangsu Key Laboratory for Advanced Metallic Materials, Southeast University, Nanjing 211189, China; 3College of Arts and Sciences, Shanghai Dianji University, Shanghai 201306, China; spring3616@126.com; 4Suzhou Institute of Biomedical Devices, Southeast University, Suzhou 215000, China

**Keywords:** pure Mg wire, recrystallization, microstructure, texture, mechanical property

## Abstract

Static recrystallization plays a key role in the fabrication of thin Mg wires as well as the mechanical properties of the final wires. The effect of annealing parameters on the evolution of the microstructures, textures and mechanical properties of cold-drawn pure Mg wire was studied by means of optical microscopy (OM), electron backscatter diffraction (EBSD), a tensile test and a hardness test. This study shows that the mechanical properties of as-annealed pure thin Mg wire is affected not only by the average grain size, but also the uniformity of the recrystallization grains, including the uniformity of grain size and crystal orientation distribution (more random texture component). With increasing annealing temperature and time, the uniformity of recrystallization grain size first improved and then declined after obvious grain growth. At the same time, the randomness of the basal texture component declined with the development of recrystallization. Annealing at 300 °C for 30 min caused the most uniform grain size and orientation distribution in the microstructures, thus contributing to the best plasticity among all experimental wires. It is reasonable to conclude that more uniform and regular recrystallized grains and a more randomly distributed crystal orientation would be benefit for the mechanical properties of Mg wires.

## 1. Introduction

Mg and its alloys have shown considerable application potential in the field of implant materials [1,2,3], owing to its high mechanical properties, good biocompatibility and biodegradability. Today, Mg alloy implantation devices have a large number of clinical needs in the field of orthopedic surgery, such as gastroenteric staple, woven vascular stent, suture wire and reinforcement of polymer matrix composite [4,5,6,7].

Among all the plastic deformation processes, cold drawing is considered the most mature and efficient method to fabricate thin metal wires. However, it is widely recognized that Mg and its alloys, with the hexagonal close-packed crystal structure, exhibit poor ductility and formability at room temperature compared with traditional fcc and bcc metals. With the recent development of research in Mg deformation, thin wires of Mg and its alloys were successfully fabricated by cold-drawing along with the subsequent static recrystallization annealing [8,9,10]. In the whole wire preparation procedure, multi-pass cold drawing and annealing treatment are performed repeatedly and the diameter of the wires gradually decreases. One purpose of annealing is to optimize the microstructure of severely cold-drawn wires to improve the deformability for the subsequent drawing to prepare fine wire. Another purpose is to modify the microstructure of the final thin wires to obtain preferable integrated mechanical properties [9].

Based on the current application limitations of Mg alloys, most of the findings [11,12,13,14] have been focused on dynamic recrystallization at high temperatures, which is obviously different from the static recrystallization mechanism realized by heat treatment after severe cold deformation. To date, there have been some papers reporting the static recrystallization process of Mg and its alloys after severe cold drawing deformation [9,15,16]. However, the mechanical properties, especially plasticity, which is particularly important for the further cold drawing of thin Mg wires, are rarely reported in the literature.

Microstructures and texture resulting from severe cold plastic deformation will inevitably affect static nucleation and recrystallization. Moreover, annealing temperature and time determine the subsequent mechanical properties and deformability of Mg wires. Our previous work [8] systematically studied the microstructures, texture and mechanical properties evolution during cold drawing. The excellent cold drawing formability indicates that pure Mg has broad prospects in the preparation of thin wires.

In this study, the static recrystallization process of cold-drawn pure Mg wire, including recrystallized grain size and uniform texture evolution with different annealing parameters, was investigated. It is crucial to provide theoretical support for the design and fabrication of micro Mg deformation profiles and medical devices.

## 2. Materials and Methods 

### 2.1. Materials

Pure Mg (>99.9%) billets were employed in the present study. The billets were first hot extruded at 650 K, reaching an extrusion ratio of ~20. The as-extruded thick Mg wires with an initial diameter of 3.0 mm were obtained as described in the paper [8]. In the present paper, the constant strain multi-pass drawing was used, namely, keeping the true stain consistent between adjacent drawing dies. The thick wires were successively cold drawn with ultimate accumulated true stain ~138%, where the fracture occurred frequently (>50%). 

The recrystallization investigation was based on cold drawn wires with a diameter of 1.5 mm. All the annealing treatments were performed in the electric resistance furnace. In order to eliminate the work hardening effect caused by cold deformation and restrain the over-grown of grains, the annealing temperature range from 150 to 350 °C and time range from 5 to 120 min were selected.

### 2.2. Methods

For the Mg wires with different annealing processes, microstructural observations were performed on the longitudinal section cut along the central axis of the wire and analyzed by an Olympus BX60M optical metallographic microscope (Olympus, Tokyo, Japan). The samples for OM analysis were mechanically polished and then etched with acetic picral (0.84 g picric acid, 2 mL acetic acid, 7 mL H_2_O, 14 mL ethanol) for 3–5 s. The grain size was counted through the lineal intercept procedure according to the ASTM E112-96 (2004) standard. 

The crystallographic orientation distribution, and misorientation angle distributions of the longitudinal section of wires were examined using electron backscatter diffraction (EBSD) in a scanning electron microscope (SEM) (FEI, Eindhoven, The Netherlands) with the field emission gun operating at 20 kV. The EBSD samples were firstly carefully mechanically sanded with a series of SiC (400#, 1200#, 2000#, 3000# and 5000#), followed by silica suspension (OPS) polishing. Then, electrochemical polishing was carried out in a solution of 5% natal acid in ethanol at 20 V for ~30 s with temperature ranging from −20 to 0 °C. The samples were then quickly rinsed with ethanol and dried.

The mechanical properties were evaluated by a CMT5105 electronic universal testing machine (Sans, Shenzhen, China) with the tensile axis parallel to the extrusion direction. The gauge (L) of the wires was 100 mm in length and the strain rate for tensile testing was 1 mm/min. After fracture, the length (L’) of each tested wire within the gauge was carefully measured. The final calculated elongation percentage is defined as (L’-L)/L. The Vickers hardness (HV) test was performed on FM-700 microhardness tester (Future-Tech, Kawasaki, Japan). The test surface was cut along the central axis of the wire and mechanically polished. The load of the hardness test was 300 g with a dwell time of 10 s. For each sample, 10 points were tested to calculate the average value. In order to be more precise, the highest and lowest values were excluded from the calculation. 

The recrystallization fraction was measured by metallographic microstructure study. The microstructure at different annealing temperatures and times can be divided into recrystallized or non-recrystallized regions. By taking the recrystallized fraction with respect to the total area of the region, the fraction of recrystallization was calculated. The complete recrystallization is defined when the recrystallization fraction reached 95%.

## 3. Results

In contrast to dynamic recrystallization of Mg alloys during plastic deformation at high temperature, static recrystallization through heat treatment is essential to obtain a subsequent deformation capacity. Based on the previous work, the annealing temperatures of Mg wires with an ultimate deformation of 138% were chosen at 150 °C, 200 °C, 250 °C, 300 °C and 350 °C to investigate the recrystallization performance.

### 3.1. Microstructure 

The metallographic structure of the as-drawn and as-annealed wires is shown in Figure 1 and Figure 2, respectively. There was fibrous deformed structure without any visible equiaxed grain in the as-drawn sample. After annealed at 150 °C for 5 min and 30 min (Figure 2a), the microstructures were similar to those of the as-drawn specimen which consisted of typical deformation microstructures as well as a few irregular recrystallized areas with some curved grain boundaries. With annealing time prolonged or temperature increased, small amounts of tiny and near equiaxed grains appeared, as marked by the arrows in Figure 2a,b. However, owing to the large number of non-uniform microstructures and bending grain boundaries, recrystallization only partially occurred at 150 °C and 200 °C even after 120 min. After annealed at 250 °C for 5 min, the recrystallization fraction was significantly increased. With the annealing time prolonged to 30 min (Figure 2c), the recrystallization was almost completed and the average grain size was 8.3 µm, although the recrystallized grains were non-uniform. As the annealing time was increased to 120 min, grain growth occurred, where the average grain size reached 13.8 µm. Most of the recrystallized grains became equiaxed with flat boundaries at 250 °C/120 min. With the increase of temperature, the recrystallization occurred more rapidly. A faster complete recrystallization within a relatively short time (5 min) occurred when annealed at 350 °C. Moreover, there was a rapid grain growth with the extension of holding time at 350 °C. 

### 3.2. Mechanical Properties

In order to further investigate the recrystallization of Mg, the mechanical properties after annealing were also examined. The microhardness values of Mg wires as a function of annealing time are plotted in Figure 3. It can be observed that the annealing temperature had a more significant effect on the hardness values than the holding time. Moreover, all of the curves show similar changing trends as the following two stages: (1) a sharp decline in 5–15 min annealing, (2) a stable region from 15 to 120 min. 

The tensile properties of the Mg wires with different annealing treatments are plotted in Figure 4. The strain–stress curves of the samples are provided in Appendix A. After annealing at lower temperatures (150 °C and 200 °C), the Mg wires exhibited relatively high yield strength and low plasticity. With the temperature increased to 250 °C, there was an evident decline of yield strength after 120 min. Meanwhile, the elongation was improved to 9% within a short holding time (5 min). Higher temperatures (300 °C and 350 °C) resulted in a more serious and rapid decline of yield strength. In comparison, the elongation evolution was quite different at higher temperatures, which first increased to the peak value and then declined as time passed.

According to the metallographic microstructure during annealing, the tensile curves of the specimens at different recrystallization stages are shown in Figure 5. The plastic stage of the wire annealed at 200 °C/5 min was quite short, indicating a rapid fracture and poor plasticity of Mg wires. With the development of recrystallization, there was an obvious improvement of plasticity, along with a decline of yield strength (300 °C/30 min). For the specimen annealed at a higher temperature (350 °C/30 min), both the yield strength and plasticity declined evidently. The tensile properties of the thinner wires with a diameter of 0.2 mm were also tested, where the wires annealed at 300 °C/30 min also showed best integrated mechanical properties (Appendix A).

### 3.3. Texture

Another reasonable explanation is that the recrystallization texture formed during annealing should not be ignored. It is reported that texture plays an important role in the mechanical properties of Mg [17,18]. Based on the study on the microstructure and mechanical properties during annealing, the texture evolution of Mg wires at different recrystallization stages was investigated. The measurements were carried out on the planes parallel to the drawing direction with the test areas in the central region of the samples. 

EBSD maps of Mg wires annealed at 200 °C/5 min, 300 °C/30 min and 350 °C/120 min are plotted in Figure 6a–c, respectively. In the incomplete recrystallization stage, there was still a deformation microstructure in the Mg wires, along with large numbers of irregularly shaped grains and wavy grain boundaries, as shown in Figure 6a. It is also shown that there are black areas in Figure 6a, and this is attributed to the relatively higher residual stress in the incomplete recrystallization stage, thus, a lower confident index and resolution. With the development of recrystallization, the ratio of equiaxed grains increased and the microstructure became more uniform, as shown in Figure 6b. 

With the development of recrystallization, the c-axis of grains tends to be more concentrated in the basal pole figures in Figure 7. The maximum pole intensities of the (0002) plane first underwent a decline and then slightly increased. Upon annealing at 350 °C for 120 min, there was a basal texture strengthening in the Mg wire. It is also shown that the texture component of <11$\bar{2}$0> parallel to the drawing direction weakened with the development of recrystallization. Although the analysis of texture based on the EBSD images was not representative of the whole material, the significant evolution of texture indicates that there was selective grain growth during recrystallization. 

Moreover, all of the misorientation angle distributions give priority to high angle boundaries with a weak peak ~30°. This corresponds to the position of $\left\{10\bar{1}1\right\}-\left\{11\bar{2}0\right\}$ double twins in Mg [17]. Both of the ratios of low angle (<10°) distribution of grains and the peak ~30° were relatively high in the specimen annealed at 200 °C/5 min, as seen Figure 8a. It is indicated that the recrystallization was not complete and there was a higher fraction of twins, both of which were not beneficial to the plastic deformation of Mg at 200 °C/5 min. At the complete recrystallization stage, the ratio of angle distributions below 30° of specimen annealed at 350 °C/120 min was obviously higher than that annealed at 300 °C/30 min, corresponding to the concentration of the basal texture component in Figure 6c. 

## 4. Discussion

During the annealing of Mg wires, recovery and recrystallization occurred, resulting in an obvious evolution of microstructure and mechanical properties. After annealing at 150 °C, there was a visible decline in microhardness of Mg, especially when the annealing time was longer than 15 min. The softening behavior during this stage is due to the dislocations rearranging into lower energy configurations [19]. Meanwhile, the metallographic microstructure of Mg annealed at 150 °C within 30 min was similar with the as-drawn specimen with very few visible recrystallized grains, indicating that the deformed samples underwent recovery. When the temperature was increased to 200 °C, the metallographic microstructure was obviously different from the as-drawn sample, with the formation of more visible recrystallized grains. Moreover, there was a more evident decline of hardness value of the annealed Mg within 5 min, indicating a faster onset of recrystallization at a higher temperature. Although the samples annealed at 150 °C and 200 °C maintained a high yield strength, the poor plasticity made it difficult to carry out the subsequent cold deformation, as seen in Appendix A. 

The annealed microstructure is important for the subsequent cold drawing to prepare thin wires. In order to quantify the recrystallization uniformity, over 200 intercepts are counted by the lineal intercept procedure according to ASTME112-96(2004) standard, as shown in Figure 9. The coefficient of variation (CV), the ratio of standard deviation and the average grain size can be used to eliminate the influence caused by the effect of different average grain size. The important values of the tensile test and the CV values of specimens at the incomplete recrystallization stage, complete recrystallization stage and grain growth stage are listed in Table 1.

The wires exhibited relatively high yield strength after annealing at 200 °C/5 min, while there was no evident improvement of elongation (1.6%), compared with that of the cold-drawn Mg wires (2.1%), which was reported in our previous work [8]. With the development of recrystallization, there was a continuously declining yield strength in the annealed Mg. This is attributed to the increasing average grain size of samples during annealing. At the complete recrystallization stage (300 °C/15 min), the Mg wires obtained a better elongation of 8.1%. The plasticity was further promoted to 13.9% as the annealing time prolonged to 30 min, although the grains were coarser (9.8 µm). The similar phenomenon was observed in samples annealed at 250 °C, where the wires with coarser grains exhibited better plasticity after being annealed at 250 °C/120 min than that at 250 °C/60 min.

Generally, materials with finer grains show better plasticity at room temperature [20]. In the present work, the improvement of elongation at 300 °C/30 min and 250 °C/120 min cannot be satisfactorily explained by the effect of average grain size. This indicates that in addition to grain size, some other key factors should be considered in the plastic deformation of Mg. According to the present tensile testing results, the recrystallization microstructure can modify the mechanical properties in two other aspects: the uniformity of the annealed grains, and the texture after recrystallization [21,22,23].

Although the grains were fine after being annealed at 200 °C/5 min, there was poor uniformity of the grains at the incomplete recrystallization stage, resulting in a high CV value (0.65). It was reported that the uniform grains were beneficial to the mechanical properties of polycrystalline metals due to the better coordination of deformation in different grains [24,25]. The metallographic microstructure shown in Figure 2b also indicates that the grains have not reached an equilibrium state, namely, these grains show a more obvious non-equiaxial morphology with a large number of curved grain boundaries. As a result, the non-uniform grains cannot provide homogeneous plastic deformation during tensile tests. Likewise, when the recrystallization was just completed (250 °C/60 min and 300 °C/15 min), the grains were less uniform, as shown in Appendix A. By extending the holding time, the CV values decreased to 0.35 and 0.29 at 250 °C/120 min and 300 °C/30 min, respectively. Thus, the improvement of uniformity and regularity of the recrystallized grains contributed to a better subsequent cold deformation, despite the coarser grains.

As the recrystallization proceeded, there were some changes in the texture of Mg. It is worth noting that there was a more random crystal orientation distribution of grains after being annealed at 300 °C/30 min. The tested area was composed of grains of different colors, as shown in Figure 6b. The random crystal orientation distribution provides a more dispersed slip system, which is beneficial for coordinating the plastic deformation. Thus, it is understandable that with the randomly distributed crystal orientation, the Mg wires show better plasticity after being annealed at 300 °C/30 min. With the further development of recrystallization, the tested area in the wire was mainly composed of grains in red (Figure 6c). This can be ascribed to the consumption of small grains by larger ones with preferred orientation during annealing [26,27]. The results in Figure 7 demonstrate that the c-axis became more concentrated during annealing, which further confirms the randomness of the crystal orientation distribution that decreased during annealing. This concentration of the crystal orientation makes it hard to coordinate the deformation in different grains, thus contributing to a poor plasticity as the recrystallization further developed. 

Based on the results in the present work, both the grain size and the uniformity of the grains should be taken into consideration in relation to the effect of the annealed microstructure on the mechanical properties of the Mg wires. The uniformity of the grains includes grain size uniformity and crystallization orientation uniformity. It is reasonable to expect that Mg wires with more uniform grains, including grain size uniformity and the randomness of the crystal orientation distribution, will exhibit good integrated mechanical properties, especially plasticity, which will be helpful for the further production of Mg thin wire by cold drawing. 

## 5. Conclusions

In this study, we conducted annealing on cold-drawn Mg wires. The microstructure evolution and its effect on the mechanical properties of Mg wires were investigated. It was found that the mechanical properties of Mg wire are affected not only by the average grain size but also by the uniformity of the recrystallization grains, including the uniformity of grain size and crystal orientation distribution. The uniformity of recrystallization grain size first improved and then declined after obvious grain growth, although the average grain size had a continuous increase during annealing. At the same time, the basal texture intensity first declined and then slightly strengthened. The randomness of the basal texture component declined with the development of recrystallization. Annealing at 300 °C for 30 min caused the most uniform grain size and orientation distribution in the microstructures, thus contributing to the best plasticity (13.9%) among all experimental wires. It is expected that more uniform and regular recrystallized grains and a more random distributed crystal orientation will be beneficial for the integrated mechanical properties of Mg wires.

## Figures and Tables

**Figure 1 materials-13-00427-f001:**
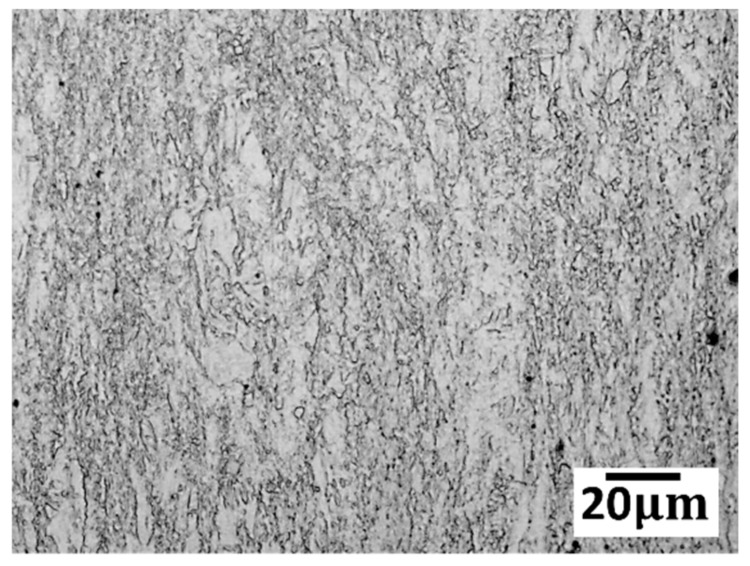
Microstructure of cold drawn Mg wire with a true strain of 138%.

**Figure 2 materials-13-00427-f002:**
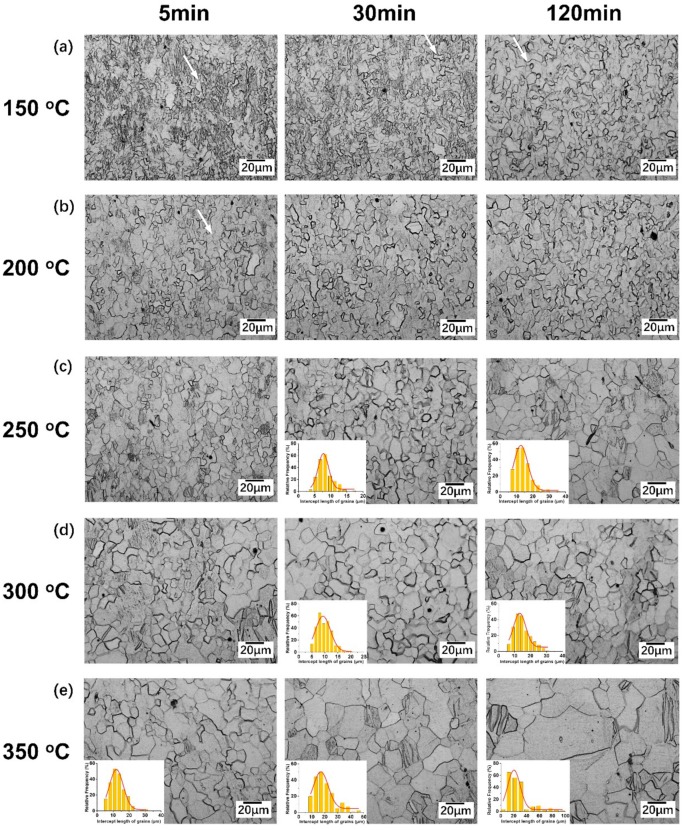
Annealed microstructures of pure Mg with different annealing temperatures after 5, 30 and 120 min. (**a**) 150 °C; (**b**) 200 °C; (**c**) 250 °C; (**d**) 300 °C; (**e**) 350 °C. The grain intercepts (size) distribution histograms of these completely recrystallized samples are shown at the bottom left.

**Figure 3 materials-13-00427-f003:**
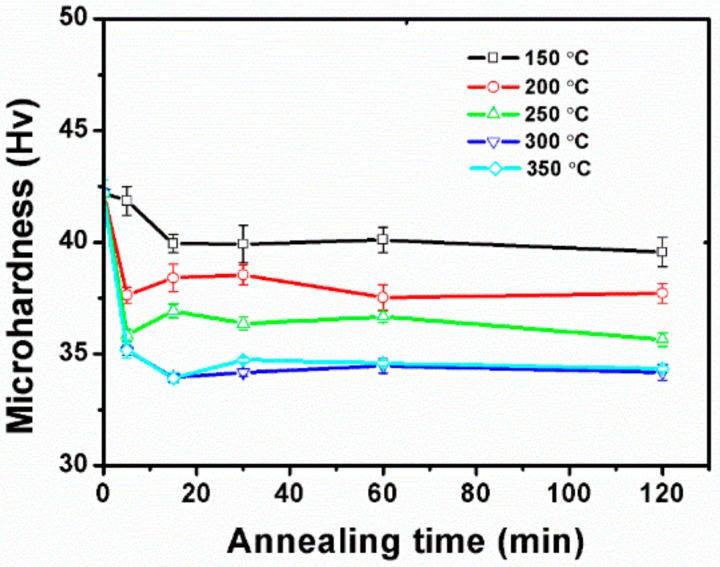
Microhardness of pure Mg with different annealing treatments.

**Figure 4 materials-13-00427-f004:**
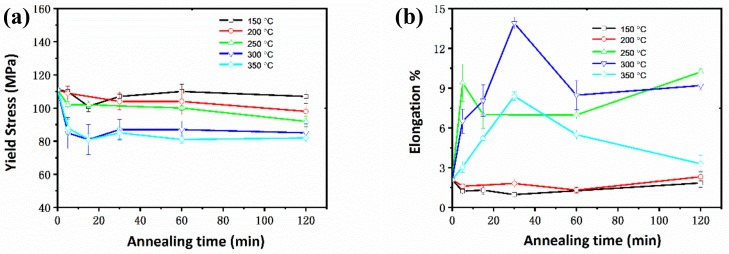
Tensile properties of pure Mg with different annealing treatments: (**a**) Yield strength; (**b**) Elongation.

**Figure 5 materials-13-00427-f005:**
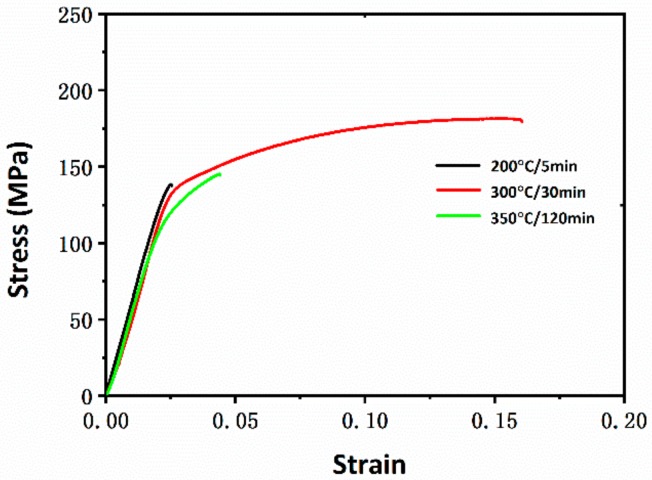
Strain–stress curves of Mg wires at 200 °C/5 min, 300 °C/30 min and 350 °C/120 min.

**Figure 6 materials-13-00427-f006:**
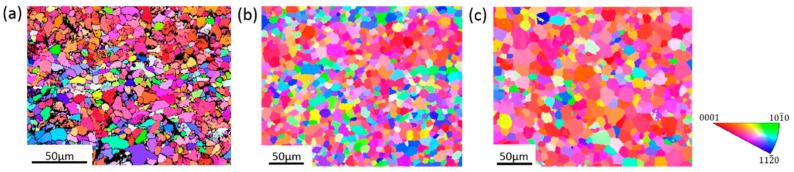
Inverse pole figure (IPF) maps of annealed Mg wires. (**a**) 200 °C/5 min; (**b**) 300 °C/30 min; (**c**) 350 °C/120 min.

**Figure 7 materials-13-00427-f007:**
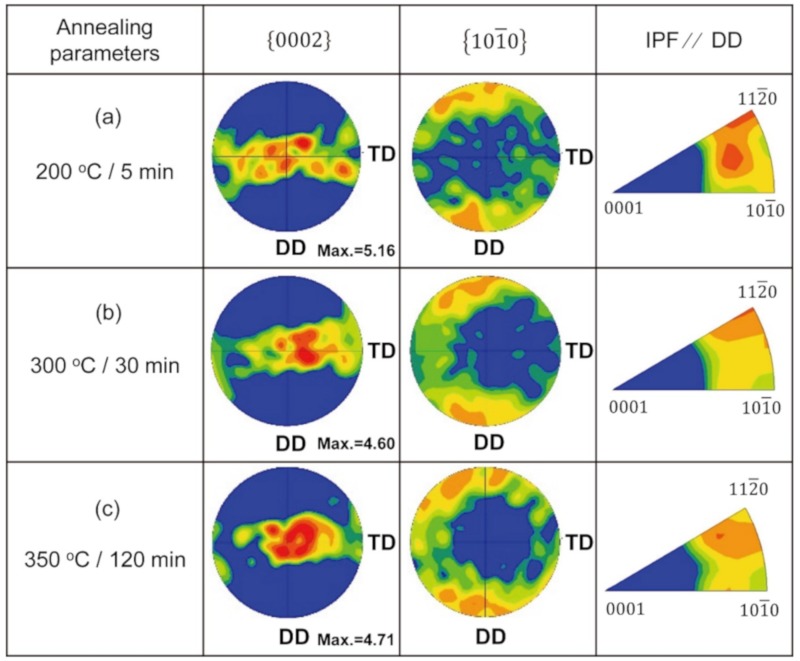
Texture evolution showing PF and IPF of pure Mg wire under different annealing treatments.

**Figure 8 materials-13-00427-f008:**
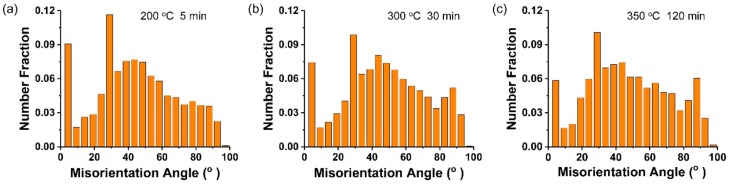
Misorientation angle distributions for annealed Mg wires. (**a**) 200 °C/5 min; (**b**) 300 °C/30 min; (**c**) 350 °C/120 min.

**Figure 9 materials-13-00427-f009:**
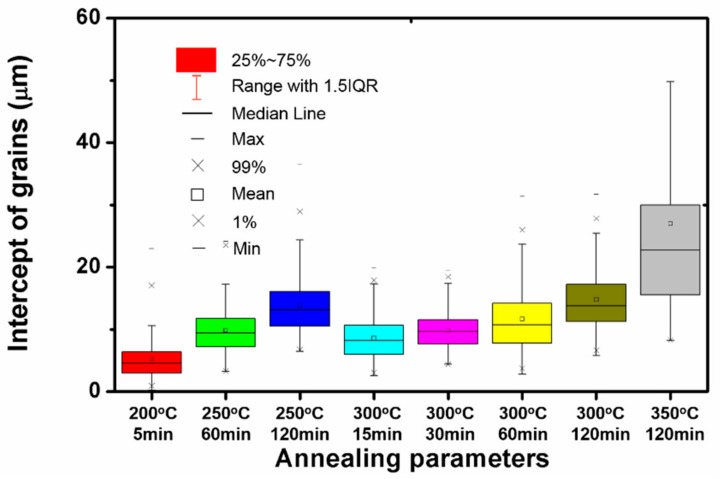
The grain intercepts distribution of the as-annealed Mg wires.

**Table 1 materials-13-00427-t001:** The statistic results of the as-annealed pure Mg wires at different recrystallization stages.

Annealing Process	Average Grain Size (μm)	Standard Deviation	Coefficient Variation (CV)	Ultimate Strength (MPa)	Yield Strength (MPa)	Elongation (%)
200 °C/5 min	5.3	3.43	0.65	140	112	1.6
250 °C/60 min	9.9	3.87	0.39	166	100	7.0
250 °C/120 min	13.8	4.80	0.35	177	93	10.2
300 °C/15 min	8.6	3.41	0.40	167	91	8.1
300 °C/30 min	9.8	2.88	0.29	181	90	13.9
300 °C/60 min	11.7	5.17	0.44	167	91	8.5
300 °C/120 min	14.8	4.98	0.34	174	89	9.2
350 °C/120 min	27.0	16.98	0.63	147	81	3.3

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
