# Peer review of "Evolution of Recrystallized Grain and Texture of Cold-Drawn Pure Mg Wire and Their Effect on Mechanical Properties"

_materials, 2020, doi:10.3390/ma13020427_

Round 1

Reviewer 1 Report

This manuscript deals with static recrystallization process and its impact on the deformability of the cold-drawn Mg wire. The authors present the evolution of the microstructure, texture, and mechanical properties with respect to annealing condition. However, the present reviewer unfortunately fines little contributions to advance scientific novelty in this study. Furthermore, the manuscript contains some issues which make it impossible to accept it for publication in Materials, in its present form.

1. Best check for a huge number of basic grammar flaws, typos, etc.
For example, (1) 'KJ/mol' should be changed into 'kJ/mol', (2) Must put a space between a number and its unit, etc. (3) Must use one temperature unit between 'K' or 'degree celsius'

2. For microhardness test, how many tests has been conducted for each sample? Though we can roughly check the tendency of the results, the authors must include error range in Fig. 2.

3. For tensile test, the reviewer cannot understand why the authors didn't include the stress-strain curves of each sample. Furthermore, how was strain measured for tensile testing?

4. For Figs. 4(a) and 6(a), the authors mentioned the presence of tensile twins according to the misorientation angle distribution in Fig. 6(a). However, the reviewer cannot find the trace of tensile twins in Fig. 4(a). The author may be able to improve the quality of EBSD image.

5. For recrystallization kinetics, It is very well known that the relationship among recrystallization kinetics, annealing T, and time. How was the recrystallization fraction measured and which criteria was used to determine recrystallization or non-recrystallization?

6. Overall the present results in the manuscript are quite predictable (recrystallization kinetics, hardness evolution with respect to annealing condition). The reviewer expect that the authors may improve the discussion about the mechanical properties more logically based on the metallurgical aspects, including the stress-strain curves.

Reviewer 2 Report

This paper presents the recrystallization behavior and mechanical properties of cold-drawn pure Mg wire. Although the paper contains some interesting results, the paper cannot be published at this stage. In the following the main three reasons for this decision are listed as well as some minor comments and corrections for the authors if they wish to resubmit the manuscript.

1) The novelty of the present study is unclear. In the “Introduction”, the authors describe that there have been only a few papers reporting the “static” recrystallization behavior of Mg and its alloys. On the other hand, the authors referred some previous literatures of the static recrystallization behavior of Mg and its alloys (Ref; 15, 18, 19, 22, 23). In particular, the analysis method in this paper seems to be quite similar to that in Ref 15. Thus, the authors should insist the novelty of the present study by comparing to the previous reports of “static” recrystallization behavior in Mg and its alloys.

2) The detail analysis of recrystallization behavior seems to be lack in this paper. First of all, No recovery process is considered in this paper. The relationship between recovery and recrystallization is very important to investigate the recrystallization behavior, but it is not discussed. For instance, the softening in Fig. 2 may correspond to the recovery and/or recrystallization processes, but the relationship between the hardness and recovery and/or recrystallization is unclear. Secondly, the recrystallization process during annealing at each temperature is unclear. For instance, the recrystallization process of the specimens annealed at 250 and 300 °C is shown in Fig. 7, but that of the specimens annealed at 150, 200, and 350 °C is not shown. Why did not the authors estimate the value of n in the specimens annealed at 150, 200, and 350 °C? The authors should perform the more detail analysis of recrystallization behavior.

3) The conclusions are unsuitable. The authors insist that the recrystallization kinetics during annealing is well described by the JMAK model, but is it the most important finding in this paper? It appears that the main purpose of the present study is to investigate the static recrystallization behavior and mechanical properties experimentally. In the “Introduction”, the theoretical analysis by the JMAK model is not mentioned. Therefore, the purpose and conclusions in the present study are inconsistent.

Minor comment:

1) The authors often use the unsuitable English for the technical paper. For example, “But the structure…” (Page 3, Line 104) and “…shouldn’t be ignored” (Page 5, Line 141). The authors should correct “But” and “shouldn’t” to “However” and “should not”, respectively.

2) The authors describe that recrystallization only partially occurs at 150 °C and 200 °C, with a few fine recrystallized grains around the deformed structure (Page 3, Line 97-98). However, I cannot recognize the recrystallized grains in Fig. 1. The authors should show the recrystallized grains in Fig. 1 with an arrow.

3) In the section of 3.1, the authors show the “approximate” grain size in each specimen. However, the authors can estimate the precise average value of the grain size from the histogram in Fig. 1. Thus, the authors should show the precise average value of the grain size in each specimen.

4) In the section of 3.2, the specimens annealed at 250 °C/5 min, 300 °C/30 min, and 350 °C/120 min are chosen. On the other hand, in the section of 3.3, the specimens annealed at “200” °C/5 min, 300 °C/30 min, and 350 °C/120 min are chosen. Why did the authors investigate the texture of the specimen annealed at “200” °C/5 min?

5) The authors insist that the lower exponent is due to the non-random recrystallization site present in the material (Page 8, Line 195). Su et al. [22] also suggested that another reason for the deviation of n is due to the growth rate of nuclei not being a constant. In addition, it is reported that the value of Avrami exponent can be affected by many factors. The authors should analyze the meaning of Avrami exponent obtained in this paper.

6) Was the activation energy (91 kJ/mol) calculated from only the data annealed at 250 and 300 °C? In that case, the value of activation energy may not be reliable.

7) What is the “main” reason for the low elongation of specimen annealed at 350 °C/120 min? In other words, which effect is greater, grain size or texture?

Round 2

Reviewer 1 Report

Most of the issues raised in the review have been addressed. 

Some more concerns are point out below to improve the updated paper. Once the below new issues are resolved, the manuscript may be acceptable for publication in Materials. 

comment #1

Grammar errors and typos in the revised manuscript should also be carefully checked.

Comment #2

In Fig. 5, the authors added the tensile test results. Figure caption should be revised and the sample name in Fig .5(a) should be corrected. 

Comment #3

Last paragraph in Introduction section, "It is crucial to provide theoretical support for the design and fabrication of the micro Mg deformation profiles and medical devices." 

In the manuscript, what is the theoretical support? The authors may mention some more discussion in a scientific manner with theoretical support. 

Reviewer 2 Report

This paper presents the recrystallization behavior and mechanical properties of cold-drawn pure Mg wire and contains some interesting results, but a series of mandatory corrections are suggested in the following:

1) According to the authors’ comment, it is difficult to estimate the fraction of recrystallization at lower temperature. How was the recrystallization fraction of 250 and 300°C- annealed specimens estimated? It is certainly difficult to estimate the recrystallization fraction from the optical micrographs, but it can be estimated by KAM analysis of EBSD measurement data. Moreover, how did the authors determine whether the recrystallization was completed or not? The authors should show the method for quantifying the recrystallization behavior.

2) Page 3, Line 128-129. The authors describe that the annealing time had less influence on the annealed microstructure when the temperature was below 350°C. In other words, is it true that the annealing time at 150, 200, 250, and 300°C had less influence on the annealed microstructure? I cannot agree with the authors’ opinion.

3) In Fig. 5 (bar graph), the annealing parameters are “250”°C/5 min, 300°C/30 min, and 350°C/120 min. On the other hand, the annealing parameters are “200”°C/5 min, 300°C/30 min, and 350°C/120 min in Fig. 5 (S-S curve). Which is correct? Furthermore, the data in Fig. 4(b), Fig. 5 (bar graph), and Table 1 is inconsistent. For instance, elongation of specimen annealed at 350°C for 120 min in Fig. 5 (bar graph) is approximately 8%, whereas that in Fig. 4(b) and Table 1 is approximately 3-4%. The authors should explain the inconsistency.

4) Page 7, Line 223. The authors describe “In the incomplete recrystallization stage (Fig. 4(a))”, but is it Fig. 6(a)?

5) Page 11, Line 319-320. The authors describe “annealed at 250°C/5 min”, but is it 200°C/5 min?

6) Page 11, Line 348 and 350. The authors should correct “haven’t” and “can’t” to “have not” and “cannot”, respectively.

7) Page 11, Line 346-355. I understand the authors’ argument. However, there is no evidence that the improvement of uniformity and regularity of the recrystallized grains contributed to a better subsequent cold deformation. At least, the authors should discuss the relationship between the uniformity and regularity of the recrystallized grains and mechanical properties by referring some relevant previous papers (not limited to Mg and its alloys).
